Cluster and trajectory analysis of motivation in an emergency remote programming course

Jahr Andres 1
Meza Javiera 1
Munoz-Gama Jorge 1
Herskovic Luis 2
Herskovic Valeria 1 vherskovic@uc.cl
1 Department of Computer Science, Pontificia Universidad Católica de Chile , Santiago , Chile
2 Escuela de Gobierno, Universidad Adolfo Ibáñez , Santiago , Chile
Vassileva Julita
Electronic publication date: 2024 Jan 12
Publication date: 2024
Volume: 10
Electronic Location ID: e1787
Received 2023 Jul 26; Accepted 2023 Dec 8
Copyright: © 2024 Jahr et al.
Copyright year: 2024
Copyright holder: Jahr et al.
License: This is an open access article distributed under the terms of the Creative Commons Attribution License, which permits unrestricted use, distribution, reproduction and adaptation in any medium and for any purpose provided that it is properly attributed. For attribution, the original author(s), title, publication source (PeerJ Computer Science) and either DOI or URL of the article must be cited.
License URL: https://creativecommons.org/licenses/by/4.0/

Keywords: Computer education, Emergency remote course, Programming, Motivation

Funding: ANID FONDECYT 1220202 ANID FONDECYT 1211210 IDeA I+D 2210048 National Center for Artificial Intelligence CENIA FB210017 Basal ANID This work was partially funded by ANID FONDECYT 1220202, ANID FONDECYT 1211210, IDeA I+D 2210048, and National Center for Artificial Intelligence CENIA FB210017, Basal ANID. The funders had no role in study design, data collection and analysis, decision to publish, or preparation of the manuscript.

==============================
Emergency remote teaching is a temporary change in the way education occurs, whereby an educational system unexpectedly becomes entirely remote. This article analyzes the motivation of students undertaking a university course over one semester of emergency remote teaching in the context of the COVID-19 pandemic. University students undertaking a programming course were surveyed three times during one semester, about motivation and COVID concern. This work explores which student motivation profiles existed, how motivation evolved, and whether concern about the pandemic was a factor affecting motivation throughout the course. The most adaptive profile was highly motivated, more prepared and less frustrated by the conditions of the course. However, this cluster experienced the highest levels of COVID-19 concern. The least adaptive cluster behaved as a mirror image of the most adaptive cluster. Clear differences were found between the clusters that showed the most and least concern about COVID-19.

Introduction

Motivation, or “the energization and direction of behavior” (Elliot & Covington, 2001), can affect student learning and performance in class (Vu et al., 2021). Indeed, motivation has often been the subject of analysis in education, both in school (Ng, Wang & Liu, 2015) and university (Çebi & Güyer, 2020), as well as in multiple delivery formats, e.g., face-to-face (Ng, 2016), online (Ferrer et al., 2020), and blended (Li & Tsai, 2017). The onset of the COVID-19 pandemic forced many institutions to begin emergency remote teaching (ERT) (Hodges et al., 2020). ERT is a temporary change in the way education occurs during exceptional crisis circumstances, whereby an educational system that was previously based on face-to-face or blended teaching becomes entirely remote. Unlike online learning, in which education is planned from the outset to operate in this format, ERT is applied in a disruptive way, and is likely to return to its original mode once the emergency is over (Hodges et al., 2020).

In an ERT course, teachers and students face several challenges. For teachers, these challenges arise from the need to replan entire courses in a short period of time, acquire new competences and learn how to use new technologies (Jimoyiannis, Koukis & Tsiotakis, 2021). Students may face concentration-related problems, as well as difficulty in interacting directly with teachers (Shim & Lee, 2020) and forming bonds with their peers (Ferri, Grifoni & Guzzo, 2020). Remote working conditions may not be optimal for either group, e.g., due to cramped working spaces, unstable internet connections or having to share a computer (Oliveira et al., 2021). For example, lower access to broadband internet was found to correlate with lower engagement among students during the pandemic (Mac Domhnaill, Mohan & McCoy, 2021).

Consequently, maintaining motivation in an ERT course may be even more difficult than in a face-to-face, online or blended course. During the COVID-19 pandemic, several researchers sought to understand the complex interrelations between motivation and other factors, as well as to how to engage and motivate students during this period of involuntary online education. A systematic review of research conducted during 2020 found that a sense of ownership and accessibility were the primary factors influencing motivation (Mohtar & Yunus, 2022). The variables of self-regulated learning and first-time online learning experiences were found to be predictors of motivation (Li et al., 2022). Educational practices also can affect motivation, as one study found that the lack of effective educational practices led to low motivation in students (Teodorescu, Aivaz & Amalfi, 2022).

Although previous studies have found variables affecting motivation, these relationships are still under study, and there is a lack of research in online course motivation in the developing world, where online instruction was scarce before the COVID-19 pandemic (Teodorescu, Aivaz & Amalfi, 2022). Furthermore, although most previous studies are quantitative in nature (Mohtar & Yunus, 2022), they have studied motivation at one point in time during a semester, usually towards the end of it (Teodorescu, Aivaz & Amalfi, 2022; Li et al., 2022; Huang & Wang, 2023; Collazos et al., 2021), while the study presented in this article aims to understand whether and how motivation can vary during the semester, and how students’ concerns about COVID-19 can affect it.

Providing greater understanding of motivation in an ERT setting may help education authorities make improved decisions in the face of multiple and fluid motivational scenarios in a future crisis involving ERT. The research questions that guided this study were the following: RQ1: Which student motivation profiles exist in emergency remote courses during the COVID-19 pandemic?

RQ2: How does motivation evolve among students undertaking an emergency remote course during the COVID-19 pandemic?

RQ3: Is concern about the pandemic a factor that affects the evolution of motivation?

To answer these questions, we surveyed university students from an emergency remote programming course (ERPC) in Santiago, Chile, during the second academic semester of 2020 (August–December). Three surveys were conducted: the first was carried out in September 2020 (T1), the second in October 2020 (T2) and the third in November 2020 (T3). To quantify student motivation, the Motivated Strategies for Learning Questionnaire (MSLQ) (Pintrich et al., 1991; Credé & Phillips, 2011) was applied. This questionnaire is in the public domain, and no specific permissions are required for it to be applied (Duncan & McKeachie, 2005). Clustering was used to generate motivation profiles, and the profile of each student was evaluated across the three stages of the survey, thus enabling course-based motivation trajectories to be generated. Therefore, our study contributes to the literature by providing a dynamic view of motivation in an emergency remote course during the COVID-19 pandemic, also especially focusing on how concern about COVID affects motivation.

Emergency remote teaching and motivation

A decrease in student motivation was found during the period of the pandemic in which students switched to the ERT learning modality (Camacho-Zuñiga et al., 2021). Students experienced greater problems completing assignments on time, managing their time, organizing their studies, interacting with teachers and teaching assistants (Pelikan et al., 2021), as well as more distractions and lower levels of concentration (Hussein et al., 2020). Social and peer interaction also became a challenge for students (Pelikan et al., 2021), and in some cases led to a lack of motivation to study (Kapasia et al., 2020). Accordingly, higher levels of motivation were associated with assignments that specifically promoted social interaction (Ismailov & Ono, 2021). For example, in the case of novice programmers in a hybrid course, students were found to need a sense of belonging and connectedness to others, to maintain motivation (Lohiniva & Isomöttönen, 2021).

Emotional and mental health problems were also reported, with several studies finding experiences of stress, anxiety, depression, feelings of being overwhelmed (Camacho-Zuñiga et al., 2021) and worries about contracting COVID-19 (Aguilera-Hermida, 2020). In Latin America, one study found students to be concerned about boredom, slow internet connections, lack of privacy and reduced feedback (Collazos et al., 2021). Positive emotional and social engagement is conducive to learning (Zhu et al., 2023), so the ERT situation during COVID-19 negatively affected students’ education.

Despite these difficulties, students also reported some benefits of ERT, such as saving on commute time, decreasing costs (Hussein et al., 2020), having more time for hobbies, getting more sleep, and spending more time with their families (Aguilera-Hermida, 2020). Indeed, some studies have reported cases of students who maintained or even increased their levels of motivation, e.g., due to a sense of challenge to achieve personal goals (Rahiem, 2021).

Regarding computer science students, one study concluded that despite the advantages computer science students might be expected to have in an online environment, the difficulties for them were the same under the ERT modality as for other students (Toti & Alipour, 2021). Furthermore, the negative impact of ERT was found to be greater for students in earlier grades, and significant differences in terms of motivation to learn were reported when comparing undergraduate and graduate students.

Understanding motivation profiles

The Motivated Strategies for Learning Questionnaire (MSLQ) has been used to assess the motivations and learning strategies of students (Credé & Phillips, 2011). The present study uses the motivation section of the instrument, which consists of six scales. Although some researchers have used a variable-centered approach, i.e., analyzing each variable individually, this research uses a person-centered approach, which involves studying the person or group as a whole and measuring how they behave under the combination of certain variables (Hayenga & Corpus, 2010).

Several studies have attempted to find the most adaptive and least adaptive (or maladaptive) clusters of students, whereby the greatest academic benefits are achieved by students whose profiles consist of the best combination of motivational, learning and psychological variables (Kong & Liu, 2020). Adaptive profiles are generally considered to be those that combine high levels of intrinsic goal orientation (Liu et al., 2021), self-efficacy (Kong & Liu, 2020), task value (Liu et al., 2014), control of learning beliefs (Çebi & Güyer, 2020) and low levels of anxiety (Liu et al., 2021), which is also found in courses associated with computer science (Çebi & Güyer, 2020). There is not a consensus as to whether extrinsic goal orientation is related to more or less adaptive profiles (Çebi & Güyer, 2020). More adaptive profiles tend to be associated with better results (Broadbent & Fuller-Tyszkiewicz, 2018).

Taking gender into account, one study found that the cluster with more women experienced slightly better levels of self-regulation, but that they were less prepared for the online learning environment (Yukselturk & Top, 2013). Conversely, another study reported that women were over-represented in a cluster associated with low self-efficacy; however, this could be due to the context, which was a highly competitive, male-dominated business administration degree (Bråten & Olaussen, 2005).

While previous research has infrequently studied student movements between clusters, some research considers the transition of students from one profile to another at two points during the period of study. Movements by individuals from the most adaptive to the least adaptive profile, and vice versa, are very rare (Bråten & Olaussen, 2005; Ng, Wang & Liu, 2015). Our research considers using exactly the same profiles at three set points in the semester, enabling studying student motivation trajectories.

Study context

COVID-19

The COVID-19 outbreak was declared a pandemic in March 2020 (World Health Organization, 2021). The pandemic seriously impacted people’s lives, e.g., through the direct fear of the SARS-CoV-2 virus itself, due to the possible consequences that infection might have on people or their loved ones (Günaydın, 2021). The present study was conducted before the approval and use of vaccines and drugs against COVID-19, so this type of fear may have been an important factor in shaping the emotions and decisions made by our participants. Several instruments have been developed to measure and study fear of COVID-19, e.g., the Fear of COVID-19 Scale (FCV-19S) (Ahorsu et al., 2020).

Generally, psychological distress is associated to academic burnout in higher education students (Emerson, Hair & Smith, 2022), and poorer wellbeing is correlated to decreased self-reported academic performance (Malta et al., 2022). Previous studies found a significant correlation between fear of COVID-19, depression and anxiety (Ahorsu et al., 2020), as well as anger, fear and disgust (Mailliez, Griffiths & Carre, 2021). However, fear of COVID-19 was found in one case to have a positive impact on academic motivation (Günaydın, 2021).

Our study took place in Chile. Two weeks after the first case of COVID-19 was detected in March 2020, several contingency measures were implemented, including curfews and a plan in which each geographical area of the country was assigned a set of restrictions (e.g., quarantines and the closure of non-essential establishments) (Chilean Government, 2021). In the three periods in which online surveys were conducted to collect data for this article, the average number of daily cases was T1 = 1,770 (median weekly average per 100,000 inhabitants (MWA) = 8.98); T2 = 1,378 (MWA = 7.54); and T3 = 1,303 (MWA = 7.18) (Ministry of Science, Technology, Knowledge and Innovation (Chile), 2021).

University and course

The Pontificia Universidad Católica de Chile (UC) is a prestigious university located in Latin America, with more than 3,500 teachers and 33,000 students (Quacquarelli Symonds (QS), 2020). Previous research has highlighted the differences during the ERT period between universities with established online programs, and those that had to unexpectedly switch to online learning with no prior experience (Teodorescu, Aivaz & Amalfi, 2022). UC belonged to this second group, albeit with some experience in the use of online classes, especially due to civil unrest in the country during the second semester of 2019, which forced many courses to conduct online activities for a brief period.

This study was carried out in the Introduction to Programming undergraduate course, a compulsory module at UC for engineering, economics, physics, statistics and astronomy students. It is also on a list of electives for students from certain programs and can be taken optionally by any other student. The course teaches 1,200 students per semester, divided into between eight and 10 groups.

Prior to the pandemic, assessments in this course consisted of coding assignments submitted through a virtual mailbox, as well as written exams. However, the course became an ERPC during the pandemic and adopted the use of online judges for all assessments, i.e., web-based applications that allow for the storage of programming problems in which, through predefined inputs and outputs, students’ code can be automatically evaluated (Yera et al., 2018).

Materials and Methods

Procedure

Data was collected through three voluntarily completed online surveys, using the SurveyMonkey software. Eligible participants were those enrolled in the Introduction to Programming course during the second academic semester of 2020 (August to December). There were 1,085 students enrolled in the course, distributed across eight groups. All enrolled students were sent e-mail invitations to answer the surveys (once per survey) through the Canvas LMS learning management system software. Students were told that it was a survey about motivation, and that those who responded would be compensated with a small number of points towards one of their grades. For this reason, as well as to filter out students who did not answer all three surveys, we asked students to provide an identity number. The Comité de Ética en Ciencias Sociales, Artes y Humanidades from the Pontificia Universidad Católica de Chile granted approval to carry out this study (Reference number 200417001).

Measures

Demographic and descriptive data

Demographic data, including age, gender, and ethnic background, was collected, as well as information regarding prior programming knowledge. At the end of the semester, data was collected on the pass/fail status of students and their final grade (a number from 1.0 to 7.0, with 4.0 being the minimum passing grade).

MSLQ questions

Only the motivation section of the MSLQ questionnaire was used for this study. This section consists of six scales. To obtain a score for each scale, values were averaged for each item, which were answered on a five-point Likert scale. Each scale is described below (Credé & Phillips, 2011): Intrinsic goal orientation: whether the student perceives that they are participating in the course out of curiosity, to learn and master a subject (four items).

Extrinsic goal orientation: whether the student participates in the course for rewards, grades, competition and recognition (four items).

Task value: how the student evaluates the course in terms of importance, usefulness and interest (six items).

Control of learning beliefs: learner’s belief that course results depend on their effort rather than on external factors (four items).

Self-efficacy for learning and performance: student’s expectation that they will perform well and confidence on their skills (eight items).

Test anxiety: negative thoughts that a student may experience during a test (five items).

COVID concern and emergency remote conditions

Four additional questions were included to represent emergency remote conditions. Of these, “I feel frustrated because my courses are online” used a five-point Likert scale; “How comfortable do you feel with the use of a computer in your day to day?” used a three-point Likert Scale and “My internet connection is stable” and “I have a computer for my exclusive use” had “Yes/No” options.

A COVID concern variable, with questions on a five-point Likert scale, that incorporated not only the fear of the risk of exposure to the virus (as previous scales such as FCV-19S) but also the concern regarding the risk posed by the virus to others, was also included (Table 1). (Cronbach’s alpha in T1 = 0.71; Cronbach’s alpha in T2 = 0.74; Cronbach’s alpha in T3 = 0.76).

Table 1 COVID concern items.

Variable	Items	
COVID concern	I am afraid/worried of getting infected with COVID-19	
I am afraid/worried that someone in my home will be infected with COVID-19	
I am afraid/worried that a relative or acquaintance outside my home will be infected with COVID-19	
	I find it difficult to sleep because I am worried about COVID-19	

Participants

There were 745 valid responses to survey 1, 678 valid responses to survey 2 and 727 valid responses to survey 3. The responses used for this study were only those in which respondents had provided full answers to the three surveys, in addition to having signed the consent form. Responses were also removed when participants did not provide their student ID, provided an incorrect student ID (e.g., their name, or less characters than valid IDs), or answered the survey more than once (only the first answer was considered). Complete responses were collected from 481 students (318 men, 158 women, five others; average age = 18.94, minimum age = 18 years, age SD = 2.06), i.e., 44.3% (481/1,085) of all enrolled students. Of the 481 students, 88.15% were in their first year of university, 83.58% were studying engineering, and 30.35% reported having some previous programming experience. Since each of the 481 students answered the survey three times, a universe of 1,443 responses was collected.

Analytic approach

Cluster analysis

Student profiles were differentiated using the k-means clustering technique (Jain, 2010), using the six MSLQ-motivation scales as input. Two k-means techniques (elbow and silhouette method) were used to find the number of clusters, and a dendrogram was used to validate that a reasonable number of clusters had been chosen.

Subsequently, the z-scores of the clusters in each of the scales were calculated. Scores above 0.5 were categorized as high, between 0.0 and 0.5 as moderate-high, between 0.0 and −0.5 as moderate-low, and below −0.5 as low, as in previous research (Ng, 2016).

Statistical tests were applied to evaluate significant differences between the profiles and variables considered in the study, in order to distinguish and categorize each of the clusters. In general, one-vs-all classifications were applied to find profiles with higher or lower values in their variables compared to the remaining clusters. Next, we explain each statistical test and the rationale for applying it. Chi-square test of independence: to determine the presence of dependence between the clusters and the categorical variables. This allowed us to see whether the proportion of engineering students was dependent on the cluster, as they could potentially have a higher affinity with the course.

Two proportion z-test: to compare the proportions of people trained in the categorical variables among the different profiles. We used this test to analyze whether each cluster had a higher or lower proportion of some categorical variable, e.g., gender or previous programming experience.

Kruskal-Wallis test: to contrast if a variable was equally distributed among the profiles. We analyzed students’ age, to understand whether there was a significant difference between ages in each cluster.

Mann-Whitney-Wilcoxon test, to establish whether, in general, any particular cluster had a numerical variable with higher or lower values with regards to others. We used this test to analyze whether there were differences between clusters in three variables that had answers in a Likert scale.

Welch test: to establish whether any particular cluster differed in value from another in a numerical variable, considering situations in which the variance was not homogeneous. In our study, we used this test to analyze students’ final grades, as in the case of this variable, variance was not homogeneous. This analysis allowed us to understand whether grades in each cluster were higher or lower than in the other clusters.

All tests were run at a significance level of 5%. For cases of multiple comparative tests between clusters, a Holm-Bonferroni correction was executed with an α-correction of 0.016. All the analyses were performed using Python 3.8.

Trajectory analysis

In order to visualize the trajectories of the students, we applied a process mining (van der & Wil, 2016) approach. The directly-follows graph (DFG) algorithm (van der & Wil, 2019) in its Disco software implementation (Günther & Rozinat, 2012) was used to model, by means of absolute frequencies, the number of events in each profile and the nature of the transitions between each one, as well as to reflect their respective beginning and completion points in the course. This algorithm was used because it is easily interpretable by education experts (Maldonado-Mahauad et al., 2018).

Results

This section answers the research questions posed in the Introduction. Each subsection represents one research question and its respective answer.

Motivation profiles in ERT during the COVID-19 pandemic (RQ1)

We obtained four clusters. Figure 1 shows the z-score values of each of the considered dimensions: intrinsic goal orientation, extrinsic goal orientation, task value, control of learning beliefs, self-efficacy and test anxiety. The figure shows cluster 1 and cluster 2; and cluster 3 and cluster 4 to be almost mirror images of each other.

Figure 1 Z-scores of the MSLQ scales for each cluster.

To evaluate the consistency of the MSLQ scales used in the clusters, Cronbach’s alphas were calculated for each of the three periods studied during the semester. The scales with the best consistency were self-efficacy and task value, all of the values of which were equal to or greater than 0.9. These were followed by the test anxiety and intrinsic goal orientation scales, both of which had values equal to or greater than 0.75. Finally, control of learning beliefs and extrinsic goal orientation showed less satisfactory results, with the first having an alpha lower than 0.7 at T1 while the latter had alphas lower than 0.7 at T1, T2 and T3. Table 2 summarizes the results obtained from the z-scores of the four profiles. Each cluster is described below.

Table 2 Interpretation of the z-scores of each profile.

	Cluster 1	Cluster 2	Cluster 3	Cluster 4	
Intrinsic goal orientation	Low	High	Moderate high	Moderate low	
Extrinsic goal orientation	Low	High	Low	High	
Task value	Low	High	Moderate low	Moderate high	
Control of learning beliefs	Low	High	Moderate high	Moderate high	
Self-efficacy for learning and performance	Low	High	Moderate high	Moderate low	
Test anxiety	High	Low	Low	High	
Label	Passive learner	Active learner	Indifferent learner	Reward-oriented learner	

Cluster 1 (19.68%) is notable due to all its dimensions being low except for test anxiety, which is the highest among all groups (z-score = 1.06). This group has the lowest scores in intrinsic goal orientation (z-score = −1.40), task-value (z-score = 1.50), control of learning beliefs (z-score = −1.48) and self-efficacy for learning and performance (z-score = −1.49). This cluster was assigned the name ‘Passive Learner’ as it represents the least adaptive profile and, in general, the lowest levels of motivation and self-efficacy.

In contrast, Cluster 2 (25.71%) is notable due to all its dimensions being high except for test anxiety, which is low. This group has the highest scores in intrinsic goal orientation (z-score = 1.42), task value (z-score = 1.29), control of learning beliefs (z-score = 1.33) and self-efficacy for learning and performance (z-score = 1.31). This cluster was assigned the label ‘Active Learner’, for having, in contrast to ‘Passive Learner’, the highest levels of motivation and self-efficacy, and therefore represents the most adaptive profile.

Cluster 3 (25.71%), provides the lowest scores in extrinsic goal orientation (z-score = −1.21) and test anxiety (z-score = −1.12). It also shows a low-moderate task value. The remaining dimensions are moderate-high. This cluster was assigned the label ‘Indifferent Learner’ because it yields moderate or low levels of motivation, in conjunction with the lowest anxiety.

Cluster 4 (28.89%) is notable due to having the highest score on extrinsic goal orientation (z-score = 1.23). Its scores on test anxiety are also high and on both task value and control learning beliefs, moderate-high. However, its intrinsic goal orientation and self-efficacy for learning and performance scores are moderate-low. This cluster was assigned the label ‘Reward-oriented Learner’, as it had the highest value in extrinsic goal orientation, in addition to high anxiety.

To characterize each of the clusters, a descriptive analysis was conducted. Table 3 shows the results of each variable in a differentiated manner, segmented by profile and total study population. Table 4 displays the statistical tests performed on each of the variables, and summarizes the main results.

Table 3 Descriptive statistics of the variables, segmented by cluster.

Variable	Passive learner	Active learner	Indifferent learner	Reward-oriented learner	Total population	
Participants	284	371	371	417	1,443	
Age						
Mean	18.95	18.96	19.00	18.85	18.94	
SD	1.12	2.35	3.01	0.92	2.06	
Gender						
Male	139 (48.04%)	202 (78.71%)	265 (71.43%)	258 (61.87%)	954 (66.11%)	
Female	145 (51.06%)	73 (19.68%)	105 (28.30%)	151 (36.21%)	474 (32.85%)	
Other	0	3 (0.81%)	0	3 (0.72%)	6 (0.42%)	
Prefer not to say	0	3 (0.81%)	1 (0.27%)	5 (1.20%)	9 (0.62%)	
Engineering students	229 (80.63%)	314 (84.64%)	316 (85.18%)	347 (83.21%)	1,206 (83.58%)	
Have previous programming experience: yes	37 (13.03%)	174 (46.90%)	120 (32.35%)	107 (25.66%)	438 (30.35%)	
Status						
Pass	242 (85.21%)	357 (96.23%)	349 (94.07%)	393 (94.24%)	1,341 (92.93%)	
Fail	17 (5.99%)	6 (1.62%)	4 (1.08%)	6 (1.44%)	33 (2.29%)	
Plagiarism	5 (1.76%)	5 (1.35%)	10 (2.70%)	10 (2.39%)	30 (2.08%)	
Withdraw	20 (7.04%)	3 (0.81%)	8 (2.16%)	8 (1.92%)	39 (2.70%)	
Final grade (1–7)						
Mean	5.23	6.58	6.07	6.00	6.01	
SD	1.29	0.95	1.28	1.21	1.27	
Frustration due to online course (1–5)						
Mean	3.81	2.98	3.17	3.66	3.39	
SD	1.16	1.28	1.31	1.30	1.32	
Exclusive computer use: yes	270 (95.07%)	365 (98.38%)	361 (97.30%)	409 (98.08%)	1,405 (97.37%)	
Comfort with computer						
Comfortable	44 (15.49%)	196 (52.83%)	149 (40.16%)	133 (31.89%)	522 (36.17%)	
Uncomfortable	78 (27.46%)	165 (44.47%)	195 (52.56%)	242 (58.03%)	764 (52.95%)	
Neither	162 (57.04%)	10 (2.70%)	27 (7.28%)	42 (10.07%)	157 (10.88%)	
Stable connection: yes	219 (77.11%)	330 (88.95%)	332 (89.49%)	338 (81.06%)	1,219 (84.48%)	
COVID concern (1–5)						
Mean	2.89	4.25	3.63	3.74	3.68	
SD	0.85	0.86	0.75	0.72	0.91	

Table 4 Summary of applied statistical tests.

Variable	Test	Result	
Age	Kruskal-Wallis test	No significant differences between the clusters.	
Gender	Two proportion z-test	Passive learner has a higher proportion of women than the other clusters.	
Active learner has a lower proportion of women than the other clusters.	
Engineering student	Chi-square test	The proportion of engineers is independent of the cluster.	
Previous programming experience	Two proportion z-test	Passive learner has a lower proportion of students with previous programming experience than the other clusters.	
Active learner has a higher proportion of students with previous programming experience than the other clusters.	
Pass/fail status	Two proportion z-test	Passive learner has a lower proportion of approved students than the other clusters.	
Final grades	Welch test	Passive learner has lower grades than the other clusters.	
Active learner has higher grades than the other clusters.	
Frustration because the courses are online	Mann-Whitney-Wilcoxon test	Active learner has less frustration because the courses are online than the other clusters.	
Exclusive computer	Two proportion z-test	No significant differences between the clusters.	
Comfort with daily computer use	Mann-Whitney-Wilcoxon test	Passive learner has less comfort with daily computer use than the other clusters.	
Active learner has more comfort with daily computer use than the other clusters.	
Stable connection	Two proportion z-test	No significant differences between the clusters.	
COVID concern	Mann-Whitney-Wilcoxon test	Passive learner has less concern about COVID than the other clusters.	
Active learner has more concern about COVID than the other clusters.	

Representation of trajectories (RQ2)

Each of the 481 students answered the survey three times, so we had a total of 1,443 responses. The four clusters were created from all the data. Then, we returned to our initial data, in which for each student, we had their survey answers at three points in time. The three surveys were then classified as pertaining to one of the clusters, so for each student, we had their trajectory from a first, to a second, to a final third profile.

Individual students’ motivation trajectories are shown in Fig. 2, as obtained from our data by using the Disco process mining software. It displays, as rectangles, each of the four types of student profiles. The arrows between them represent transitions between the profiles, and the numbers represent the students who followed the corresponding trajectory. The inverted triangle at the top represents the beginning of the semester, while the outgoing arcs show the number of students who began the course in each respective profile (T1). For example, 71 students began as ‘Passive Learners’, while 166 students began as ‘Indifferent Learners’. Similarly, the rectangle towards the bottom represents the end of the academic semester (T3). Its input arcs show the number of students who completed the semester in each profile—e.g., 92 students completed the semester as ‘Indifferent Learners’. Accordingly, the intermediate arcs represent the movements between profiles throughout the semester (T1 to T2 and T2 to T3), e.g., five students moved from ‘Active Learners’ to ‘Passive Learners’ during the semester. In addition, under the name of each profile is the absolute frequency of events in each cluster, i.e., the number of students that constitute a profile.

Figure 2 Individual students’ motivation trajectories.

Beginning and end of the trajectories

Most of the students began the semester in the ‘Indifferent Learner’ profile, with 166/481 (34.51%) cases, while only 71/481 (14.76%) began in the ‘Passive Learner’ profile. However, the profile in which the least number of learners completed the semester was ‘Indifferent Learner’, with 92/481 (19.12%) cases. The rest of the clusters showed no major differences at T3.

Comparing the initial and final status of each profile, the ‘Indifferent Learner’ cluster was that with the greatest outflow of students. At the end of the academic period, it consisted of 44.57% fewer people than at the beginning. Conversely, the remaining clusters showed an increase of students. The profile that experienced the highest increase of students was ‘Passive Learner’, which grew by 81.69%.

Movements between profiles

Regarding movements between profiles, the largest number was from the ‘Indifferent Learner’ cluster to the ‘Reward-oriented Learner’ cluster with 73 cases, followed by movements from the ‘Reward-oriented Learner’ cluster to the ‘Passive Learner’ cluster with 55 cases, then from ‘Indifferent Learner’ to ‘Passive Learner’ with 45 cases, and from ‘Reward-oriented Learner’ to ‘Indifferent Learner’ with 43 cases.

The ‘Indifferent Learner’ profile had the highest outflow with 158/371 cases, i.e., 42.58% of those who began the semester in the profile. Similarly, the ‘Reward-oriented Learner’ profile also experienced a high degree of outflow, with 129/417 cases of students moving to another cluster, or 30.93%. The ‘Reward-oriented Learner’ cluster was also the group that experienced the highest inflow, with 135/417 cases, thus representing 32.37% of its total case frequency.

Most frequent variants of the trajectories

Table 5 shows the 10 most frequent variants of the student trajectories. The Type column shows the movement patterns followed regarding the three periods studied during the semester.

Table 5 The 10 most frequent variants of the trajectories.

Type	Frecuency	T1	T2	T3	
AAA	70 (14.55%)	Active learner	Active learner	Active learner	
AAA	43 (8.94%)	Indifferent learner	Indifferent learner	Indifferent learner	
AAA	42 (8.73%)	Reward-oriented learner	Reward-oriented learner	Reward-oriented learner	
AAA	39 (8.11%)	Passive learner	Passive learner	Passive learner	
ABB	23 (4.78%)	Indifferent learner	Reward-oriented learner	Reward-oriented learner	
ABB	22 (4.57%)	Indifferent learner	Active learner	Active learner	
AAB	15 (3.12%)	Reward-oriented learner	Reward-oriented learner	Passive learner	
ABB	14 (2.91%)	Indifferent learner	Passive learner	Passive learner	
ABB	14 (2.91%)	Reward-oriented learner	Passive learner	Passive learner	
ABB	11 (2.11%)	Active learner	Reward-oriented learner	Reward-oriented learner	

The first four variants are of the AAA type, i.e., in 40.33% of cases, students remained in the same profile throughout the three periods in question. However, the next 6/10 most frequent variants, totalling 20.58% of all cases, are primarily of the ABB type, except for one AAB type. If all trajectories (51) are considered, in 63.61% the profile at T2 and T3 remained the same for the student. This includes the AAA and ABB patterns.

There were no cases of movements from the ‘Passive Learner’ to the ‘Active Learner’ profile and only five cases in which a student moved directly from the ‘Active Learner’ to the ‘Passive Learner’ profile (Fig. 2). Considering all trajectories, there was only one case of a student who began the semester in the ‘Passive Learner’ profile and completed it in the ‘Active Learner’ equivalent, and only three cases of students who began in the ‘Active Learner’ and finished in the ‘Passive Learner’ cluster.

Comparison of motivational trajectories according to COVID concern (RQ3)

A comparative study was conducted between the trajectories of students with a high COVID concern z-score (greater than 0.5) and those with a low z-score (less than −0.5), based on an average taken from across T1, T2 and T3. Figure 3 shows the trajectories of students with high z-scores for COVID concern, corresponding to 166/481 cases, or 34.51% of the total population. The predominant profile here was ‘Active Learner’, as 75/166 (45.18%) students began in this cluster while 101/166 (60.84%) completed the semester in this cluster. For the ‘Passive Learner’ profile, 5/166 (3.01%) students began the semester in the cluster and 3/166 (1.80%) students ended the semester in it. Only 5/166 (3.01%) students with a high COVID concern moved towards this profile throughout the entire academic period.

Figure 3 Trajectories for students with COVID concern z-score higher than 0.5.

Figure 4 shows the behavior of students with a low COVID concern. This segment was composed of 150/481 (31.18%) students. The ‘Passive Learner’ profile predominated, having the highest inflow of cases. Moreover, 93/150 (62.00%) students completed the semester in this profile. In the case of the ‘Active Learner’ cluster, only 12/150 (8.00%) students began the semester in this cluster and even fewer ended the semester in this cluster (5/150, 3.33%).

Figure 4 Trajectories for students with COVID concern z-score lower than −0.5.

Discussion

This study found that motivation varies during an ERT programming course, which is in line with our expectation that maintaining motivation during such courses may be difficult. Although previous researchers have sought to untangle the relationship between motivation and other factors, usually at the end of the semester, our study (1) presents student profiles in an understudied population (students during the pandemic at a university in South America), (2) provides understanding on how motivation can vary during the semester, and (3) provides insights about how concern about COVID-19 and motivation are related. In the next two sections, we discuss our results regarding student profiles and student trajectories, highlighting how they contribute to the literature and finally discussing two implications of our findings.

Student profiles

Student final grades are in line with previously identified positive correlations between intrinsic goal orientation (Broadbent & Fuller-Tyszkiewicz, 2018), self-efficacy (Valle et al., 2015), task value (Ng, 2016) and academic performance. One of the causes that could be a contributing factor to these differences between the profiles is the learning environment and psychological preparation for the ERT scenario. The ‘Active Learner’ profile showed lower levels of frustration due to the course being online, which could indicate that this profile was less affected from ERT conditions. In contrast to the ‘Passive Learner’ profile, the ‘Active Learner’ cluster had students who were more comfortable using a computer and had more previous programming experience. For the ‘Passive Learner’ cluster, this would be a disadvantage not only for being in a programming course, but also for being in an online environment.

The ‘Active Learner’ profile was the one that showed the highest levels of COVID concern. Although several studies have shown that fear of COVID is associated with adverse psychological outcomes, some previous research has also found COVID fear to have a positive impact on academic motivation (Günaydın, 2021). In that study, researchers found that students’ problem-solving skills were a positive and significant predictor of COVID-19 fear, theorizing that people who are more rational are more realistic and may, therefore, have a greater concern about the virus (Günaydın, 2021). Our research contributes to the literature by showing that the trajectories of students with high and low COVID concern are different; and that students with high COVID concern are more likely associated to more adaptive profiles, while those with low COVID concern are more likely associated to less adaptive profiles.

When comparing the ‘Indifferent Learner’ and ‘Reward-oriented Learner’ profiles, it was difficult to determine which of the two was the more adaptive. Both had fairly similar values in terms of grades and MSLQ variables but differed significantly in extrinsic goal orientation and test anxiety. Previous studies have reported varying results regarding the role of extrinsic goal orientation (Çebi & Güyer, 2020; Hayenga & Corpus, 2010), however, anxiety has generally been associated with negative motivational profiles (Ng, 2016). Regarding the descriptive variables, although no significant differences with the other clusters were detected, there are some differences: the ‘Indifferent Learner’ cluster showed a slightly higher proportion of students with previous programming experience, somewhat lower levels of frustration because the courses were online, and a marginally higher proportion of students who felt comfortable using a computer. This suggests that these students were a little more adjusted to the course and modality. Given the above, we posit that ‘Indifferent Learner’ students were more adaptive than ‘Reward-oriented Learner’ students. This research contributes to the literature by identifying these four profiles and arguing they are, from most to least adaptive: ‘Active Learner’, ‘Indifferent Learner’, ‘Reward-oriented Learner’, ‘Passive Learner’.

Student trajectories

This research has provided a method to understand how student motivation changes throughout a semester. Student trajectories showed a predominance of the ‘Indifferent Learner’ profile at the beginning of the semester. However, it was the cluster with the least number of students to complete the semester and the only one which experienced a decrease over the semester. This indicates that a significant proportion of the students initially had more moderate levels of motivation, COVID concern, and anxiety, but subsequently evolved over the course of the semester towards less conservative positions. Naturally, this evolution may be tied to the evolution of the pandemic during the time period of this study. The ‘Passive Learner’ profile was the group with the fewest students at the beginning of the course, but which experienced the highest growth. This suggests high levels of lack of motivation, increased anxiety, and a lack of COVID concern. On the other hand, the ‘Reward-oriented Learner’ profile experienced the highest absolute frequency of learners. It was the profile with the highest inflow and the second highest outflow, being a high turnover cluster. Moreover, 50.93% of all students spent at least one period of the semester in this group, while 27.02% spent two or more periods therein. This indicates that a considerable proportion of the students perceived, at some point during the semester, that their predominant motivation was extrinsic, while their test anxiety was high.

Regarding the variants of the trajectories, the most frequent were found to be those in which students remained in the same profile during the three periods. Specifically in this study, the ‘Active Learner’ profile was the most stable in terms of case inflow/outflow during the semester. The remaining principal variants were mostly cases in which the student began in one cluster but then moved to another and remained in that cluster during the second and third periods. This may suggest that the motivation of students in the first period was subject to expectations driven by uncertainty about the new modality and the pandemic. Another phenomenon was that there was almost no direct movement from the ‘Passive Learner’ to the ‘Active Learner’ profile and from the ‘Active Learner’ cluster to the ‘Passive Learner’ cluster. This may indicate that, despite the variability of movement between profiles, moving from a more adaptive to a less adaptive profile, and vice versa, requires a far more drastic change in terms of motivation, study habits and study environment, the occurrence of which is far less likely. This effect has also been found in other research that analyzed profile changes over time (Bråten & Olaussen, 2005).

Specifically in the trajectories with the highest and lowest COVID concern levels, substantial differences were obtained, especially regarding the ‘Passive Learner’ and ‘Active Learner’ profiles, showing that very different trajectories are possible albeit dependent on COVID concern.

Implications of our findings

We would like to discuss two implications of our findings, which we believe are relevant for future researchers as well as practitioners: (1) being aware of motivation fluctuations, and (2) the successful use of process mining as a method of analysis of motivation trajectories.

Awareness of motivation fluctuations

During the semester, 59.67% of the students experienced at least one movement from one motivation profile to another, although movements between the most different profiles (‘Passive Learner’ and ‘Active Learner’) were rare. As we only measured motivation at three points in the semester, we posit that students may very often fluctuate between motivation profiles, e.g., feeling in some weeks an intrinsic desire to learn while in other weeks (perhaps when test dates approach) feeling higher test anxiety. We believe instructors need to be cognizant of these fluctuations, to e.g., remind students of the importance of learning rather than test scores, as well as concentrate on providing effective instructional practices.

Use of process mining to analyze motivation trajectories

This research provides a first use of process mining to analyze motivation trajectories. We found the process mining tool to be valuable in providing visualizations of student trajectories, allowing us to explore not only the general movement of students between motivation profiles, but also allowing us to understand how particular groups of students (e.g., those with higher and lower COVID concern) behaved. We believe process mining can be a useful tool so researchers with similar datasets can explore their data and gain insights from students' experience during a course or semester.

Conclusions

Four learner profiles were identified among students taking an ERT university programming course. The ‘Passive Learner’ cluster was the least adaptive, with low levels of intrinsic goal orientation, task value, control of learning beliefs and self-efficacy, and high test anxiety. This cluster had a higher proportion of women than men (51.06%), a lower proportion of students with prior programming experience, a higher proportion who felt less comfortable using a computer, lower academic results, and a lower level of COVID concern. The ‘Active Learner’ cluster was the most adaptive, an approximate mirror image of the ‘Passive Learner’ profile, with the best academic results, a significantly lower proportion of women (19.68%), and a significantly higher proportion of students with previous programming experience. However, students in this cluster had higher levels of COVID concern. The ‘Indifferent Learner’ cluster had the lowest levels of extrinsic goal orientation and test anxiety, which suggests low concern about academic performance. Finally, the ‘Reward-oriented Learner’ cluster had the highest level of extrinsic goal orientation, as well as high anxiety, thus having the highest levels of motivation to achieve improved grades and external recognition. The ‘Indifferent Learner’ profile was determined to be more adaptive than the ‘Reward-oriented Learner’ profile.

Most of the students began the semester in the ‘Indifferent Learner’ profile. As the semester progressed, some students migrated to alternative profiles, but most remained at the same profile or only transitioned between profiles once, so there was a certain stability in the levels of motivation and anxiety. The trajectories of the students who were most concerned about COVID-19 were dominated by the ‘Active Learner’ profile, while the trajectories of those least concerned related primarily to the ‘Passive Learner’ profile. This shows that the trajectories not only differ, but also are mainly composed of the most and least adaptive profiles. Differences in student motivation are not only static, as determined by the different clusters, but also dynamic, as presented by the different trajectories.

There are several limitations to this study that we would like to acknowledge. First, Cronbach’s alpha revealed low consistency in the responses associated with two scales. The surveys include a non-response bias due to self-selection. Attrition bias must also be taken into account due to participants who did not answer all three surveys. Surveys answered in an erroneous way, e.g., individuals who miswrote their identity number or who did not give informed consent, were excluded, thus possibly generating a selection bias. It is also important to consider differences in motivation that may have been generated between the eight different groups undertaking the course. Indeed, recorded classes with better quality teachers produce improved student outcomes (Clark et al., 2021). Although all teachers adhered to the same assessments and assignments, their teaching styles may have varied, potentially affecting students’ performance and motivation. Regarding study participants, a large number of students passed the course, when compared to previous semesters with pen-and-paper assessments, in which the passing rate was around 75–80%, and compared to the first post-pandemic semester that used online judges, in which the passing rate was 47%. Plagiarism detection was used during the semester in question, but it is possible that some students did not work individually and were not detected. Finally, and also regarding study participants, our sample considered students from one university and one course, who are mostly homogeneous in terms of culture, ages, and study program, which limits generalization to other populations.

As future work, consideration should be given to the inclusion of the resource management strategy questions from the learning strategies subsection of the MSLQ, which may help characterize student behavior. It is also important to study the relationship between students’ mental health and their motivation trajectories. It would also be interesting to compare motivation during the same course in a semester with in-person or online classes post-pandemic, to further understand how the pandemic and COVID-concern affect students’ behavior and motivation. Finally, we plan to expand this method to other courses and other universities, to help improve the generalizability of our findings.

Additional Information and Declarations

Competing Interests

Author Contributions

Ethics

Data Availability

The authors declare that they have no competing interests.

Andres Jahr conceived and designed the experiments, performed the experiments, analyzed the data, performed the computation work, prepared figures and/or tables, authored or reviewed drafts of the article, and approved the final draft.

Javiera Meza conceived and designed the experiments, performed the experiments, authored or reviewed drafts of the article, and approved the final draft.

Jorge Munoz-Gama conceived and designed the experiments, analyzed the data, prepared figures and/or tables, authored or reviewed drafts of the article, and approved the final draft.

Luis Herskovic analyzed the data, authored or reviewed drafts of the article, and approved the final draft.

Valeria Herskovic conceived and designed the experiments, analyzed the data, prepared figures and/or tables, authored or reviewed drafts of the article, and approved the final draft.

The following information was supplied relating to ethical approvals (i.e., approving body and any reference numbers):

The Comité de Ética en Ciencias Sociales, Artes y Humanidades from the Pontificia Universidad Católica de Chile granted approval to carry out this study (Ethical Application Ref: 200417001).

The following information was supplied regarding data availability:

The data is available at GitHub and Zenodo:

- https://github.com/vherskovic/data_surveys_motivation.

- vherskovic. (2023). vherskovic/data_surveys_motivation: rawdata (rawdata). Zenodo. https://doi.org/10.5281/zenodo.8184149.

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
