# Peer review of "Cluster and trajectory analysis of motivation in an emergency remote programming course"

_PeerJ Computer Science, doi:10.7717/peerj-cs.1787_

## Round 0.1 · original submission · Major Revisions

The reviewers find the topic relevant and the paper generally well written. Their main concerns are the need for better positioning of the main research question and respectively - the contribution within the current literature, providing justification for the statistical analysis methods applied, and discussing the possible impact of other factors on the motivation of learners in the limitations and future work. Are the results specific to Chile or could they be generalized to other Latin American, countries?

Reviewer 1 ·

Basic reporting

The paper titled "Cluster and Trajectory Analysis of Motivation in Emergency Remote Programming Courses" aims to investigate the dynamics of student motivation in the context of emergency remote programming courses. The paper appropriately references relevant literature and provides sufficient field background and context. The inclusion of recent studies and theoretical frameworks adds depth to the paper and establishes a strong foundation for the research.

The paper adheres to a professional article structure, which includes clear sections such as Introduction, Literature Review, Methodology, Results, Discussion, and Conclusion. Additionally, it provides figures and tables that are well-organized and complement the text. The figures and tables are appropriately labeled and discussed within the text, enhancing the paper's overall quality. However, the figures are faint making it difficult to understand and some of them were not explained in detailed.

The authors did not share raw data.

The paper maintains a self-contained structure and ensures that the results are relevant to the hypotheses. The authors clearly state their research questions and hypotheses in the introduction, and the results presented in the subsequent sections directly address these questions.

The authors presentation of results aligns with the expectations of scholarly research.
The paper contributes to the understanding of student motivation in emergency remote programming courses.

Experimental design

The paper has a valuable contribution to the field, aligning with the Aims and Scope of the journal. It addresses a relevant and meaningful research question related to student motivation in emergency remote programming courses. The research question is clearly defined. However, the paper did not highlights how the study fills an identified knowledge gap in this domain and did not discuss the significance of their research.
The investigation conducted in this study adheres to ethical standard. The study design follow appropriate approach for data collection, analysis, and interpretation.
The paper does not provides sufficient detail and information regarding the analysis methods used in the study and their relevance. Analysis procedures, and statistical tools used were not properly described and this makes it difficult to understand the paper. For examples, in section Analytic approach, a two portion z-test, and Kruskal-Wallis test were used and their relevant were not explained in details. Mann-Whitney test and welch test were used. Why are the two test relevant? More details about the statistical analysis and their relevant should be provided. Furthermore, more details about trajectory analysis should be provided. what does number of events in each profile and nature of the transitions between each other means? The survey was conducted three times, how were the survey responses processed before the cluster analysis. Where the responses combined.
For result section RQ1,
I think that from the text "To evaluate the consistency " to end of the section should be presented first before the results of the clusters. Where the difference between the clusters significant and how did you test them?
For section RQ2, detailed explanation for Figure 2 should be provided to make the paper easy to understand.
Some grammatical errors need to be corrected.

Validity of the findings

While the study addresses an important topic, namely student motivation in emergency remote programming courses, it could benefit from a more explicit discussion of how the findings contribute to the existing literature and what potential implications they may have for both the academic community and practical applications. However, this was not discuss.

In the conclusion section, the authors connected their findings to the research objectives, ensuring that the conclusions remain relevant and aligned with the study's purpose. The limitations of the study are acknowledged, and the conclusions are appropriate, limiting their scope to supporting the presented results.

Reviewer 2 ·

Basic reporting

Authors depic an study exploring which student motivation profiles existed, how motivation evolved, and whether concern about the pandemic was a factor affecting motivation throughout the course in a context of Chile. Document is interesting.
It could be adequate to include a related work section. there are some studies made in Latinamerica aboutsimilar aspects. check the work of Collazos et al, Designing Online Platforms Supporting Emotions and Awareness
Its neccesary to specify the main hypotheses of the work

Experimental design

The study is relevant, and could be applied in different scenarios. It could be adequate to describe in a better manner the recruitment process

Validity of the findings

The study is relevant and appropriate considering the effect of lockdown caused by COVID-19. It couls be adequate to analyze what other aspects could influence in the motivation and user perception (culture, age, technological acceptance, etc). Improve further works

---

## Round 0.2 · accepted · Accept

The reviewers were happy with your response to their comments and have endorsed the publication of your manuscript. I concur with their recommendation.

Reviewer 1 ·

Basic reporting

The article was written in English with clear, unambiguous, technically correct text. It provided sufficient introduction and background to demonstrate how the work fits into the broader field of knowledge and relevant prior literature was appropriately referenced. The paper adheres to a professional article structure, which includes clear sections such as Introduction, Literature Review, Methodology, Results, Discussion, and Conclusion. The authors clearly stated their research questions in the introduction, and the results presented in the subsequent sections address the questions.
"MSLQ questions
231 Only the motivation section of the MSLQ questionnaire was used for this study. This section
232 consists of 6 scales. To obtain a score for each scale," I think that you should change 6 scales to 6 subscales. To obtain a score for each subscale. There are other occasions, for example, Each scale is described below - the scale should be subscale.
I assume that T1, T2, and T3 are different times; explicitly define what they stand for.

Experimental design

More details about the methodology and the analysis methods used in the study and their relevance have been added.

Validity of the findings

Discussions on the potential implications of the study findings have been added. Most of the issues in the previous version have been addressed.

Reviewer 2 ·

Basic reporting

Authors have considered previous comments

Experimental design

Authors have considered previous comments

Validity of the findings

Authors have considered previous comments

Additional comments

Authors have considered previous comments